# Efficiency of Recycled Biogas Digestates as Phosphorus Fertilizers for Maize

**Inga-Mareike Bach \*, Lisa Essich and Torsten Müller** 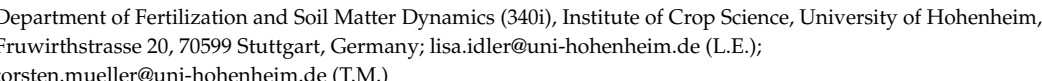

Department of Fertilization and Soil Matter Dynamics (340i), Institute of Crop Science, University of Hohenheim, Fruwirthstrasse 20, 70599 Stuttgart, Germany; lisa.idler@uni-hohenheim.de (L.E.); torsten.mueller@uni-hohenheim.de (T.M.)
\* Correspondence: IngaMareike.Bach@gmail.com; Tel.: +49-170-810-2721

**Abstract:** Despite phosphorus resources on Earth being limited, over fertilization in many agricultural situations causes significant resource consumption. Phosphorus-recycling within agricultural production can reduce global dilution into the environment and is thus essential to secure sustainable future supply. This study investigated the fertilization efficacy of phosphorus fertilizers recycled from biogas digestates in maize shoots grown under controlled greenhouse conditions, in two soils, in a pot experiment. Variables investigated were plant-available phosphorus in soil, plant biomass production, and concentration of phosphorus, calcium, and magnesium in shoots. Soils were treated with three different fertilizer fractions, separated from biogas digestates, at equivalent phosphorus concentrations, using different combinations and application techniques, isolated or in combination, and compared to triple superphosphate (TSP) as a reference. One of the fractions (P-Salt) had effects on biomass production and plant phosphorus concentration equivalent to TSP in agricultural surface soil. In the second soil (with less active soil life and nutrient content), equivalence to TSP was achieved with combinations of two recycled fractions (P-Salt and dried solids). The enhancement of the phosphorus fertilizing effect by the solids was synergistic, indicating that the solids had a soil conditioning effect. The results show that biogas digestates are a valuable source for phosphorus recycling of fractions that have equivalent or even superior fertilizing properties compared to TSP.

**Keywords:** biogas digestates; recycled phosphorus fertilizer; phosphorus fertility indicators in soil; maize

## 1. Introduction

One of the main factors increasing agricultural production is the use of organic and inorganic fertilizers. Phosphorus (P), an essential and irreplaceable element and fertilizer component, is a key factor for crop growth and quality [1]. Since exploitable P resources on Earth are limited and are, paradoxically, paired with systematic over fertilization with environmentally negative side effects, P-recycling from agricultural production is essential to secure future P supply for food production. Agriculture accounts for the consumption of approximately 80% of P from phosphate rock for fertilizer production [2]. Currently, the entire P for chemical fertilizers and feed is derived from P-rich rocks and about 70% of the known reserves are located in Morocco (Western Sahara) and China as the main exporters of P-ore [3]. Europe has no significant P mines and is highly dependent on the import of P [4]. Globally, the continued increase in population and changes in human diet will put further pressure on agricultural production to meet the growing food demand. Consequently, P is receiving more attention as a non-renewable resource [5,6]. These challenges stress the importance of finding alternative P sources in this century [7]. The requirements for alternative P-sources are multidimensional. Possible candidates must have similar characteristics compared to standard mineral fertilizers and would need to be available in large amounts with consistent quality and similar fertilization potential, as

one unique characteristic of P is its low availability in soils due to its slow diffusion and high fixation [8].

One possibility to source additional P is from recycling biogas digestates that have valuable potential as organic fertilizer or soil amendment [9] and contain high amounts of plant nutrients, including N and P [10]. Agricultural biogas production is an efficient method of converting a wide range of organic waste into biogas, heat, and residual anaerobic digestates [11]. For example, animal manure slurries are co-fermented with energy crops like maize to produce biogas [12]. The boom of the European biogas production in recent years has been a consequence of the EU Commission's initiative to limit global climate change to two degrees Celsius by changing the European energy policy [13,14], which encouraged the use of renewable resources like energy crops and livestock effluents for biogas production. According to the European Biogas Association [15], the number and electrical capacity of biogas systems in European countries increased from 6227 plants with 4158 MW capacity in 2009 to more than 17,000 plants with over 9900 MW capacity in 2017. Especially in Germany, this has led to an increased production of biogas digestates of around 65.5 Million $m^3$ $a^{-1}$ [9].

In regions with high livestock density and associated biogas production, the amounts of residual digestates are too high to be sustainably used locally without negative effects to the environment, and the handling of biogas digestates becomes an increasingly urgent and cost-intensive matter in terms of storage capacities and transport to arable farmland. Due to their high water content of around 90% [16], long-distance transport of raw digestates is neither profitable nor ecologically viable. In addition to other aspects like energy consumption, plant availability and adequate nutrient balance, one major objective of biogas digestate recycling must therefore be a drastic reduction of the water content. Due to a high proportion of organic P, plant availability of P in recovered products is often low or unpredictable, and the uptake in plants can differ substantially [17]. Johnston and Richards [18], as well as Römer [19], found that some P fertilizers from residues, such as wastewater or veal manure, ensured relatively good P availability, which indicates that the use of recycled P fertilizer in agriculture is possible.

The objective of a research project named GOBi (General Optimization of Biogas Processes) was the integral optimization of the biogas process chain to increase operational, material, energy and thereby ecological efficiency with special attention to the production of a natural, technically suitable fertilizer. As part of this, a new recycling technology for the recovery of P from biogas digestates was developed [20,21] to separate a salt containing struvite, calcium phosphate and magnesium phosphate (P-Salt) from the liquid fraction and to convert the solid digestate fraction into dried solids, resulting in nutrient-rich, storable and transportable fertilizer fractions for use as conventional fertilizers. The remaining aqueous phase was an N-rich liquid that could be used for plant irrigation without further processing. In summary, this new technology consisted of (a) mechanical separation of the digestate into a liquid and a solid phase, (b) solubilization of P from the liquid phase by addition of acid, followed by (c) precipitation of P from the liquid phase and (d) drying of the solid phase.

In the present study, the efficiency of recycled P-fertilizers, produced by the technology described above, was tested in a greenhouse pot experiment with maize plants in two different soil types, using conventional triple superphosphate (TSP) and untreated controls as reference. The first soil was a typical agricultural surface soil (0–10 cm) with active soil life and low nutrient P content compared to the second soil, a C-horizon (substratum) subsoil with less active soil life and low nutrient content, chosen as a model for less fertile soils. Maize was used as a representative, widely used input crop for biogas production. Especially at the beginning of plant growth, the P supply to maize plants is critical. *Zea mays* L. var. 'Carolinio' is a specific variety for the use as biogas substrate as it has a high dry matter yield potential. Dry matter was measured as an indicator of total biomass production of energy crops. The concentration and total content of P, Ca and Mg, as main components in the fertilizer fractions, was analyzed in plants as a measure for nutrient

uptake, and calcium–acetate–lactate extraction of phosphorus (CAL-P) was analyzed as indicator for plant-available P in soil.

P adsorption to soil particles can be greatly reduced by co-application of organic substances [8]. Muhammad et al. [22] reported that combining mineral and organic fertilizers in the form of TSP and compost led to an increased plant P availability in a greenhouse experiment with maize. It could therefore be expected that a combination of both solid fractions, the dried organic solids and the precipitated P-Salt, might result in positive effects on fertilization and soil conditioning. Therefore, a second objective was to evaluate whether the combined application of P-Salt and a dried solid fraction in different ratios (1:1 or 1:2) might influence P fertilization. Different application techniques (pre-mixed in water suspension or dry-mixed) were used to get an indication for the robustness of the combined use in agricultural practice.

Digestate drying is a commercially available technology and according to the German Biogas Association [23] 500–700 dryers are in use in Germany's biogas industry [24]. Different drying techniques with advantages and disadvantages were discussed by Salamat et al. [24]. When using different temperatures (45 °C, 70 °C and 80 °C) for the air-drying process of the solids, no significant differences regarding organic dry matter and P were observed [25], whereas a pot experiment with dried sewage sludge resulted in a decreased P availability and P-uptake in barley plants [26]. Thus, as a third objective to assess differences between the drying processes of the previously separated solid fractions, solids from two drying processes (either with a warm-air dryer at 40 °C, or with a steam-dryer at 120 °C) were compared.

Based on the objectives described above, the following hypotheses were tested:

The P-Salt alone and combinations with solids recovered from the biogas digestate have fertilizer effects on plant dry matter, nutrient concentration and on CAL-P in soil, comparable to a mineral P fertilizer (TSP).

There is a synergetic effect between P-Salt and recycled solids (air-dried and steam-dried). This effect depends on the application technique (dry or suspended) and on the ratio of P-Salt to solid fractions.

Different drying procedures of the solids (air dried solids vs. steam dried solids) lead to different fertilization effects on maize plants.

## 2. Materials and Methods

The experimental part of this study was based on a greenhouse pot experiment with maize in two different soils to assess the P fertilizing effect of three recycled P-fertilizers and their different combinations, in comparison to the reference mineral fertilizer TSP and an unfertilized control.

### 2.1. Recycled P-Fertilizers

Recycled P-fertilizers were isolated from the anaerobic digestate of a biogas plant in Kupferzell, Germany [20]. A detailed overview about the process is given in Figure 1. In a first step, the digestate was mechanically separated into a solid and a liquid fraction. Phosphate salts (P-Salt) were solubilized from the liquid phase by acidification with $H_2SO_4$, followed by precipitation by increasing the pH with NaOH. The raw digestate contained sufficient magnesium (5.5% dry matter) to allow for struvite formation. However, the ammonium–N content (2.1% dry matter) was low, and therefore less than 50% was precipitated as struvite, whereas the rest was bound as calcium phosphates and magnesium phosphates. The resulting P-Salt fraction, a mixture of struvite, calcium phosphates and magnesium phosphates, was finally dried and powdered or granulated to give a homogeneous material for dosing and fertilization. The aqueous N-rich supernatant could directly be used for crop irrigation.

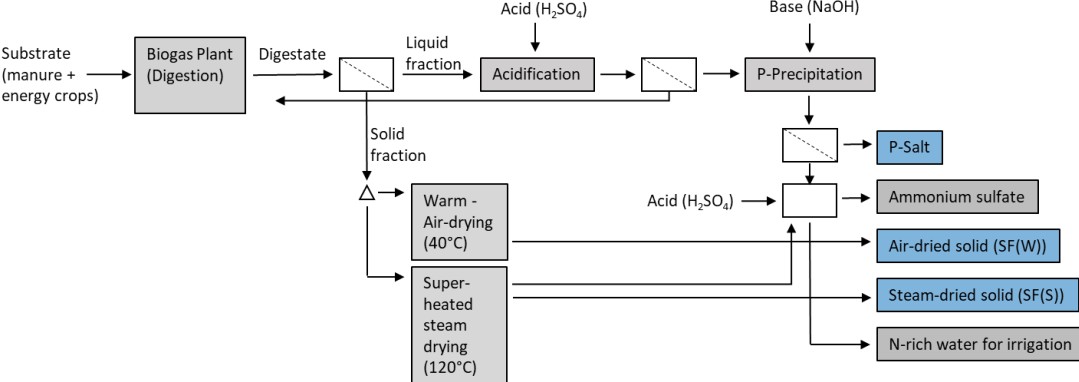

**Figure 1.** Overview of the recycling process of biogas digestates (according to [20]) to separate phosphate salts (P-Salts) from the liquid phase and to convert the solid digestate fraction into dried solids (SF (W/S) (blue boxes). All rights on this recycling process belongs to Fraunhofer IGB, Stuttgart, Germany [21].

From the solid fraction of the digestate, two types of dried solids were recovered, produced by drying to approximately 90% dry matter (DM) with either a warm-air dryer at 40 °C (air-dried separated solids, "SF(W)") or a steam-dryer at 120 °C (steam-dried separated solids, "SF(S)"). The measured P content of the different fertilizer fractions is given in Table 1. Total water-soluble P was determined by extraction with water followed by ammonium-citrate and analysis by ICP-OES according to VDLUFA II 4.1.4 [27]. A characterization of the different P forms in the recycled P fertilizers was performed using a modified Hedley-fractionation method [28,29]—for details see Wollmann et al. [30]. P, Mg, Ca, Na, K and Fe were measured using ICP-OES with $HNO_3$ extraction (DIN EN ISO 11885). Ammonium–N was determined by DIN 38406-E5.

**Table 1.** Characteristics of phosphate salt (P-Salt) and solids.

| Property/Variable | P-Salt (Precipitated from Liquid Digestate Fraction) | Steam-Dried Solids (Separated from Solid Digestate Fraction, Dried at 120 °C) | Air-Dried Solids (Separated from Solid Digestate Fraction, Dried at 40 °C) | Mineral TSP (Triple-Superphosphate, P Fertilizer as Reference) |
|---|---|---|---|---|
| Abbreviation | P-Salt | SF (S) | SF (W) | TSP |
| DM [% FM] | 69.7 | 91.6 | 95.4 | |
| Pt [% DM] | 11.3 | 2.3 | 2.1 | 19.0 |
| Water soluble P [% DM] | 0.13 | 0.35 | 0.35 | |
| Sequentially fractionated with ... in (mg P (g DM)$^{-1}$) | | | | |
| NaHCO$_3$ (easily available P) | 29.9 | 6.7 | 7.5 | |
| NaOH | 6.3 | 0.98 | 1.3 | |
| H$_2$SO$_4$ (sparingly available P) | 53.3 | 0.96 | 1.74 | |
| pH [CaCl$_2$] | 8.3 | 8.5 | 7.1 | |
| Ca [% DM] | 8.2 | 1.8 | 1.7 | 15.0 |
| Mg [% DM] | 5.5 | 0.5 | 0.5 | |
| K [% DM] | 1.1 | 1.6 | 1.8 | |
| Na [% DM] | 0.43 | 0.07 | 0.08 | |
| Fe [% DM] | 0.55 | 0.13 | 0.14 | |
| Ammonium N (NH$_4$-N) [% DM] | 2.1 | 1.4 | 1.5 | |

Dry matter (DM) in % fresh matter (FM), total P ($P_t$) content in % of DM, water soluble P in % DM and P contained in different fractions of sequentially extracted P (NaHCO$_3$, NaOH, H$_2$SO$_4$), indicated as total amounts (mg P (g DM)$^{-1}$) and pH in CaCl$_2$ and total Ca, Mg, K, Na, Fe and N in % DM of the investigated fertilizers P-Salt, SF (S) and SF (W) and the reference TSP; all% in (w/w). $P_t$ determined with dried fertilizers and aqua regia extraction, according to VDLUFA (2000) and measured with ICP-OES.

*2.2. Soil Characteristics*

The pot experiments were carried out using two soils, namely a silty loam and a clay loam, both selected for their low concentration of CAL-P, defined as P extractable in calcium–acetate–lactate, according to [31]. The silty loam was a subsoil collected at the

research station Kleinhohenheim of the University of Hohenheim, Germany, with low fertility and low microbial activity and thus not a typical, representative agricultural soil, but selected to show influences of low microbiological activity on fertilizer efficacy. The clay loam was an agricultural surface soil with high active soil life, high water retention capacity and was collected from agricultural soil in Kleinansbach, Germany.

### 2.3. Soil Sampling and Analysis of P, K, Mg, $C_{org}$, $N_t$ and pH

Soil samples were taken from each pot both before maize sowing and after harvest. Samples were stored at $-20\,^{\circ}$C until analysis. Plant available P in soil (dried at 105 $^{\circ}$C and sieved to 2 mm) was analyzed using the CAL extraction method (calcium–acetate–lactate extractable P, [31]). The same method was used for K and Mg determination. C and N contents were determined with a CN analyzer (VarioMax, Elementar Analysensysteme, Hanau, Germany). Soil pH was measured in 0.01 M $CaCl_2$ suspension [32].

### 2.4. Fertilizer Dosing

All P-fertilizers and their combinations were applied once at the beginning of the experiment to soil at a dose equivalent to 150 mg total P per kg soil (standard dose for P supply) prior to sowing of maize. In addition to the three isolated P fertilizer fractions, combinations were tested with P-Salt combined with SF (W) or SF (S) in the ratios of 1:1 or 1:2. The corresponding dose weights of single doses or combinations were calculated from actually measured P concentrations in the used fertilizer fractions. For combined fertilizers, a ratio of 1:1 means that 75 mg total P per kg soil came from the P-Salt and 75 mg total P per kg soil came from the separated solid fraction, amounting to 150 mg total P per kg soil. Accordingly, combinations of 1:2 were generated by mixing 50 mg total P per kg soil coming from the solid fraction with 100 mg total P per kg soil coming from the P-Salt. Simultaneous or sequential application was chosen in order to test P-availability effects depending on the application technique. "Dry" application was done by homogeneously mixing both components into the soil sequentially (first P-Salt, followed by SF (S/W)). For the "suspended" application, both components were first mixed together with water in a separate vessel before adding and mixing it as one product into the soil.

### 2.5. Experimental Details

Fertilization doses of the three different recycled P-fertilizers, the eight different fertilizer combinations and variants of them, as well as the reference fertilizer TSP, were given at equimolar P amounts per pot. The corresponding dose weights of single doses or combinations were calculated from P concentrations in the used fertilizer fractions pre-measured by ICP-OES (extraction with $HNO_3$). The required amounts of P-Salt, SF (W) and SF (S) and their different combinations were homogeneously mixed with 3 kg soil (dry weight) and filled into round plastic pots (4 liter volume, 20 cm diameter). The conventional fertilizer TSP was pre-mixed with deionized (DI) water to ensure the exact dosage of the small amounts and then added to the soil. Additional mineral fertilizer (ammonium nitrate ($NH_4NO_3$), potassium sulfate ($K_2SO_4$), magnesium sulfate ($MgSO_4$), Fe–Sequestren (Ferric EDTA)) was added once to each pot, including the untreated controls. After fertilization, the soil was mixed, watered to a water-holding capacity (WHC) of 70% and incubated for two days in the greenhouse at 24 $^{\circ}$C before sowing the maize seeds into the soil. The commercially used biogas crop *Zea mays* L. var. Carolinio (KWS SAAT SE & Co. KGaA, Einbeck, Germany) was used. Three seeds were sown per pot into the soil at a 4-cm depth. After germination, the number of seedlings per pot was reduced to one plant per pot. A second additional mineral fertilization event, only with N, was applied three weeks after germination. During the experiment, each pot was weight-controlled every three days and irrigated with deionized water to maintain a water content of 70% water holding capacity (WHC). A detailed overview of the conditions is included in Table 2 and in Figure 2.

**Table 2.** Experimental setup of the greenhouse pot experiments.

| | |
|---|---|
| Crop | Maize (*Zea mays* L. var. Carolinio), 3 seeds per pot; after germination reduction to 1 seedling per pot |
| Soil | Silty loam: Texture uL; pH [CaCl$_2$] 7.3; nutrient status for P, K, Mg (all CAL) in mg/kg soil: 7; 71; 210; C$_{org}$%: 0.3; N$_t$%: 0.04 Clay loam: Texture tL; pH [CaCl$_2$] 7.4; nutrient status for P, K, Mg (all CAL) in (mg (kg soil)$^{-1}$): 26; 150; 580; C$_{org}$%: 3.6; N$_t$%: 0.24 |
| Additional mineral fertilization per pot (excluding P) | Before sowing: 200 (mg N (kg soil)$^{-1}$) as NH$_4$NO$_3$, 200 (mg K (kg soil)$^{-1}$) as K$_2$SO$_4$, 100 (mg Mg (kg soil)$^{-1}$) as MgSO$_4 \cdot$7H$_2$O and 10 (mg (kg soil)$^{-1}$) Fe–Sequestren (6%) 4 weeks after sowing: 200 (mg N (kg soil)$^{-1}$) as NH$_4$NO$_3$ |
| Experimental Duration | Total: 50 days |
| Conditions | ambient greenhouse conditions (University of Hohenheim, Germany, June 2016), ca. 16 h light, 8 h dark, ca. 20 °C; initial watering to 70% water-holding capacity (WHC) with deionized (DI) water, additional watering when required (weight control every 2–3 days) |
| **P-fertilizer treatments** | |
| **Recycled P-fertilizers** | **all mg below refers to P equivalents per 1 kg dry soil** |
| P-Salt SF (W) SF (S) | 150 mg 150 mg 150 mg |
| SF (W) + P-Salt (1:1) | Dry mixed (dry): 75 mg SF (W) mixed into the soil, directly followed by 75 mg P-Salt mixed into the soil Suspended mixed (susp.): 75 mg SF (W) + 75 mg P-Salt + 50 mL DI water, pre-suspended in a separate vessel before mixing into soil |
| SF (W) + P-Salt (1:2) | Dry mixed (dry): 50 mg SF (W) mixed into the soil, directly followed by 100 mg P-Salt mixed into the soil Suspended mixed (susp.): 50 mg SF (W) + 100 mg P-Salt + 50 mL DI water, pre-suspended in a separate vessel before mixing into soil |
| SF(S) + P-Salt (1:1) | Dry mixed (dry): 75 mg SF (S) mixed into the soil, directly followed by 75 mg P-Salt mixed into the soil Suspended mixed (susp.): 75 mg SF (S) + 75 mg P-Salt + 50 mL DI water, pre-suspended in a separate vessel before mixing into soil |
| SF(S) + P-Salt (1:2) | Dry mixed (dry): 50 mg SF (S) mixed into the soil, directly followed by 100 mg P-Salt mixed into the soil Suspended mixed (susp.): 50 mg SF (S) + 100 mg P-Salt + 50 mL DI water, pre-suspended in a separate vessel before mixing into soil |
| Control treatments Triple superphosphate (TSP) Negative control | Positive reference; 150 mg DI water |

The pots were set up in a randomized complete block design on tables in the same greenhouse, using four replicates per treatment, resulting in a total of 104 pots, 52 for each soil type. C$_{org}$ = organic carbon; N$_t$ = total nitrogen.

*2.6. Maize Harvest and P, Mg and Ca Analysis*

After 50 days, the maize shoots were cut 0.5 cm above the soil surface, and fresh weight and dry weight (after drying at 60 °C for 48 h, dry matter, DM) were recorded. Analyses were carried out after microwave extraction [33] using ICP-OES (Agilent 5100, Santa Clara, CA, USA) according to DIN EN ISO 11885: dried plant material was ground using a laboratory disk mill (TS 250, Siebtechnik GmbH, Mülheim and der Ruhr, Germany) and 0.5 g of the plant material was suspended in concentrated HNO$_3$ and H$_2$O$_2$, followed by microwave extraction at 210 °C for 62 min and filtration. Shoot nutrient (P, Mg Ca) content was calculated as mg/shoot and as mg/shoot DM.

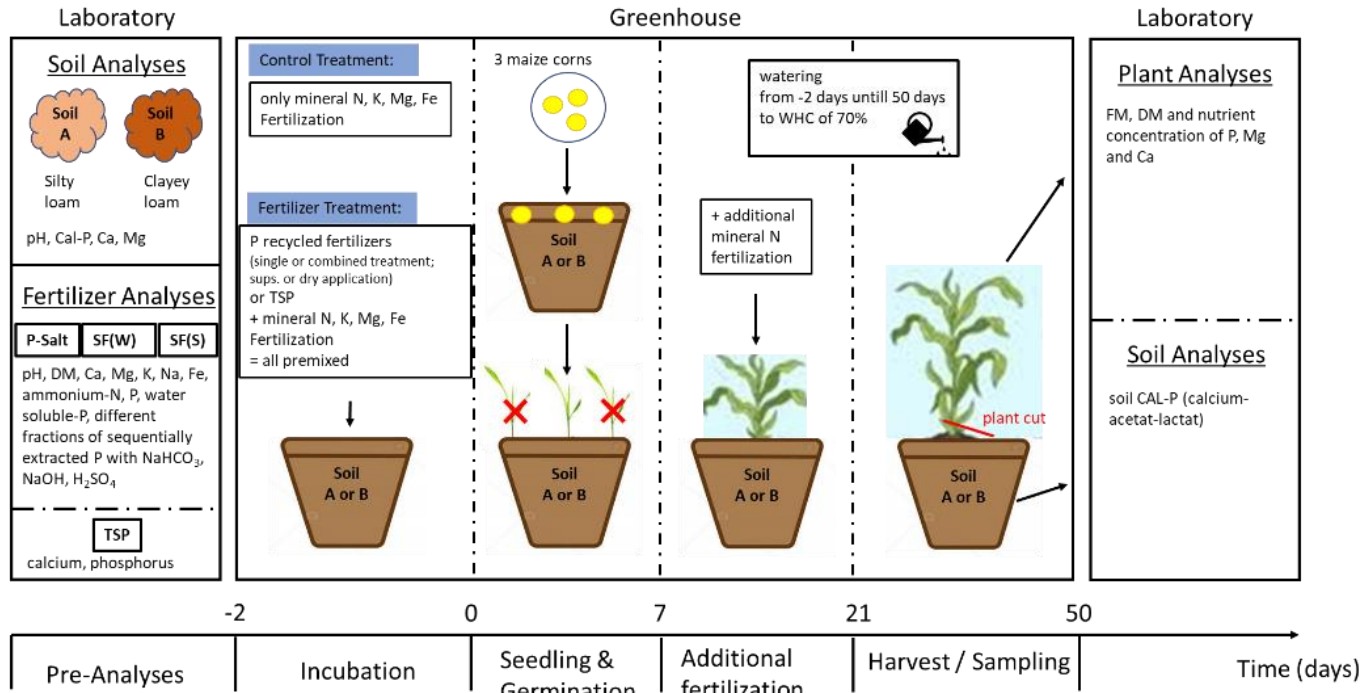

**Figure 2.** Overview of the experimental set-up of the greenhouse pot experiment with maize. Fertilizer fractions tested were P-Salt (P-enriched precipitate from the liquid fraction), SF (W) (air-dried solid fraction), SF(S) (steam-dried solid fraction), and different combinations of them, compared to unfertilized control and triple superphosphate (TSP). Maize plants were cultivated for 50 days in a silty loam and clay loam soil. DM—dry matter; FM—fresh matter; Ca—calcium; Mg—magnesium; K—potassium; N—nitrogen; Fe—iron; WHC—water holding capacity.

### 2.7. Evaluation of Synergistic Effects

In order to evaluate the effects of the combined application of P-Salt + SF (SF (W) or SF (S)), theoretical biomass production and P content was calculated from P-Salt and SF applied alone using the following equation:

Calculated DM yield for ratio 1:1 (Equation (1)) and 1:2 (Equation (2)):

$$\text{calculated DM yield} = \frac{\text{SF (W or S) DM yield} + \text{P} - \text{Salt DM yield}}{2} \tag{1}$$

$$\text{calculated DM yield} = \frac{\text{SF (W or S) DM yield} + (\text{P} - \text{Salt DM yield}) \times 2}{3} \tag{2}$$

Calculated P content for ratio 1:1 (Equation (3)) and 1:2 (Equation (4)):

$$\text{calculated P content} = \frac{(\text{SF (W or S) P content} + \text{P} - \text{Salt P content})}{2} \tag{3}$$

$$\text{calculated P content} = \frac{\text{SF (W or S)P content} + (\text{P} - \text{Salt P content}) \times 2}{3} \tag{4}$$

Measured values higher than the calculated values after combined application were interpreted as synergistic effects, whereas measured values equal to or lower than the calculated ones indicated non-synergistic effects.

### 2.8. Statistical Analysis

Data analysis was performed using the statistical computing software R version 4.0.4. A block design setup with fertilizer treatments and soil types as fixed elements and different variables (P concentration, DM yield and P content in the maize plants and CAL-P in soils) was assessed using a two-factorial linear model ($n = 4$).

The model can be described as follows:

$$y_{ijk} = \mu + \alpha_i + \beta_j + (\alpha\beta)_{ij} + b_k + e_{ijk},$$

where $y_{ijk}$ is the yield (or CAL-P or P-concentration) of the *i*th fertilizer and *j*th soil type in kth block, $\mu$ is the general mean, $\alpha_i$ is the main effect of ith fertilizer, $\beta_j$ the main effect of *j*th soil type, $(\alpha\beta)_{ij}$ the fertilizer-by-soil type interaction, $b_k$ the effect of *k*th block, and $e_{ijk}$ is the residual error. Fertilizers and soils were treated as fixed elements.

Data were log-transformed in order to meet the model assumption of normality of residuals and variance homogeneity, when necessary. Least square means and letter display for pairwise comparison were performed using the R packages emmeans [34] and multcomp [35,36]. Significance was determined at $p \leq 0.05$ using a Tukey's test, performed only on finding significant differences in the F-test. Significantly different mean values were indicated by different letters and mentioned in the text. Lowercase vs. uppercase letters were used to indicate significant differences between both soils within the same treatments, so that, e.g., the use of capital letters (regardless which one) in both soils indicates non-significance between the same treatments.

## 3. Results

### 3.1. Effect of P-Fertilizers on Biomass Yield and Plant Nutrient Concentration

Maize shoot dry matter yield (DM yield) is shown in Figure 3. In the silty loam soil (nutrient-poor subsoil, low active soil life), untreated controls had the lowest DM yield (0.9 g/plant) compared to all fertilizer treatments. When dosed alone, the increase was highest with SF (S), at a level equivalent to TSP. The recovered P-Salt alone had the lowest effect of all applied fertilizers. The highest DM yields (10.0 g/plant) were detected when SF (W or S) and P-Salt fractions were combined, in some combinations with significantly higher DM compared to the reference TSP. All combination treatments (SF + P-Salt) resulted in much higher DM yield than theoretically calculated from the single components' yields, indicating a synergistic effect between both fractions. Combinations of air-dried solids (SF(W)) with P-Salt led to higher DM yields when dosed separately, whereas those with steam-dried solids (SF (S)) gave higher DM yield when given as mixture. Mixtures with high SF(W/S) fraction content (1:1) had a higher DM yield than combinations with higher P-Salt content (1:2). Nevertheless, even the (1:2) combination of P-Salt with air-dried solids (SF(W) (applied dry) resulted in DM yield comparable to the reference TSP.

Plants grown in the clay loam soil (nutrient-rich surface soil, high active soil life) already developed a level of 5.9 g DM/plant in untreated controls, which was in the order of magnitude of the nutrient-poor subsoil after fertilization. The TSP reference dosing increased DM by only a factor of ca. 2 to 12.4 g/plant. In contrast to the silty loam, single dosing of P-Salt here had the highest effect of all treatments (14.69 mg DM yield/plant), even higher than the reference TSP. Similar to the silty loam, steam-dried solids (SF (S)) alone gave higher yields than air-dried solids (SF(W)). All combinations of the recycled fertilizer fractions were on the same order of magnitude as the TSP control. Higher P-Salt content in the combinations and SF(S) rather than SF(W) resulted in slightly higher yield. Combinations generally resulted in higher DM yield when dosed dry, however differences were less pronounced than in the silty loam. A comparison of the yields calculated from single dosing with those of the combinations showed little, mostly insignificant differences, indicating additive effects of both fraction types in this case. Overall, all treatments and combinations resulted in a significant increase of plant dry matter levels compared to untreated controls, and for both soils combinations of the recycled fertilizers were identified that gave equivalent or even higher yield than the reference TSP.

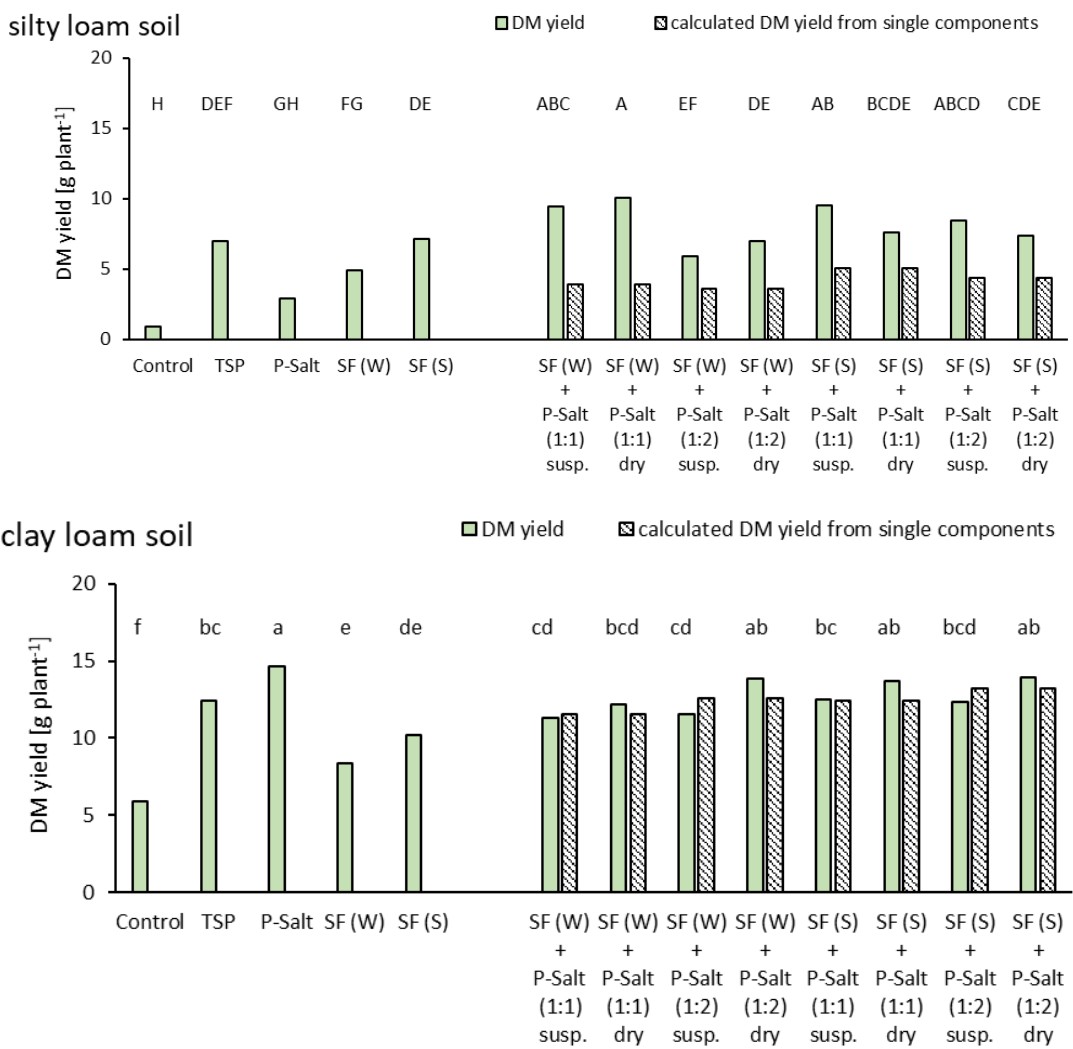

**Figure 3.** Mean shoot dry matter yield of maize (DM yield) in g plant$^{-1}$ after soil treatment with different recycled digestate fertilizers in a total dose equivalent to 150 mg P (kg soil$^{-1}$). Fertilizer fractions tested were P-Salt (P-enriched precipitate from the liquid fraction), SF (W) (air-dried solid fraction), SF(S) (steam-dried solid fraction), and different combinations of them, compared to unfertilized control and triple superphosphate (TSP). Maize plants were cultivated for 50 days in a silty loam and clay loam soil. Based on the total fertilizer dose corresponding to 150 mg P (kg soil$^{-1}$), designation 1:1 means that fertilizers were applied with a mass portion of SF (W or S) containing 75 mg P (kg soil$^{-1}$) + a mass portion of P-Salt containing 75 mg P (kg soil$^{-1}$); likewise, 1:2 means using a mass of SF (W or S) with 50 mg (kg soil$^{-1}$) + a mass of P-Salt with 100 mg (kg soil$^{-1}$). The application technique "susp." (suspended) describes that both components were pre-mixed and homogenized with 50 mL water in a vessel before being homogenized into the soil; "dry" means that the solid fraction SF was first mixed into the soil followed by the P-Salt, both without water. "Calculated DM yield from single components" represent a mathematical calculation of the combinations based on the results of the measured DM yield of each single component (P-Salt, SF(W), SF(S)) to indicate possible additive effects (material and methods: evaluation of synergistic effects Equations (1) and (2)). Data are means of four replicates each. Different letters denote significant differences (two-factorial Tukey's test with $p \leq 0.05$). Lowercase letters for the clay loam indicate that, in this case, all treatment groups were significantly different from those of the silty loam with the same treatment.

Figure 4 summarizes the nutrient concentration in mg/g DM of P, Ca and Mg in the maize shoots after harvest, as an indicator for nutrient net uptake into the plants. The P concentration in the maize shoots grown in the silty loam soil increased significantly with all fertilizer combinations and especially after treatment with P-Salt alone (10.9 mg/g DM), a combination of dry applied SF (S) + P-Salt (1:2) (5.4 mg/g DM) and SF (W) + P-Salt (1:2) (4.8 mg/g DM). These three applications resulted in much higher P concentrations in the

plants compared to the reference TSP (1.9 mg/g DM), indicating that P-mobilization from P-Salt was higher compared to TSP in this nutrient-poor soil. In contrast, the lowest, but still significant effect on P-concentration in the shoots, compared to untreated controls, was detected after single application of the solid fractions (SF (W/S) where the P-concentration was low (Table 1). In a comparison of the application technique of the combined fertilizer treatments (dry or suspended), dry application consistently resulted in much higher P concentrations compared to suspended application, especially when the ratio SF(W/S) + P-Salt was high (1:2). Even at a ratio of 1:1, higher P concentrations in the shoots were found in all dry applications compared to suspended applications. The lowest P-concentration in the shoots (2.1 mg/g DM) was found with a 1:1 suspended mixture of SF (W) + P-Salt, still significantly higher than untreated controls and not significantly different from treatment with the reference TSP.

Ca and Mg-concentration per maize plant were evaluated to monitor the influence of the P-Salt on the net-uptake of both nutrients in plants. The highest Ca concentration in the maize shoots after harvest was found in the untreated controls with 20.7 mg/g DM. All other treatments had much lower concentrations between 5.0 mg/g DM (SF(S) + P-Salt (1:1), dry) and 7.6 mg/g DM (P-Salt). Similar Ca concentrations were detected in the treatments with TSP, P-Salt and SF(W) with a single application. Combined fertilizer treatments resulted in slightly lower Ca concentrations, with a tendency of slightly higher values for higher P-Salt ratios (1:2) and using the suspended application technique vs. dry application. Overall, however, only low differences between all treatment groups were observed. The Mg concentration, like the Ca concentration, was highest in the untreated control plants (10.4 mg/g DM). All single and combined fertilizer treatments resulted in Mg concentrations of 4–5 mg/g DM, with no or only small significant differences between them. The only exception was P-Salt alone with an elevated concentration of 7.3 mg/g DM.

In the clay loam soil, the P concentration in dried maize shoots did not differ significantly between the fertilizer scenarios, even including the untreated control. None of the treatments apparently lead to an increase of P-concentration in the plants, all with levels around 2 mg/g DM. Likewise, the Ca concentration in maize shoots was undistinguishable between the fertilizer scenarios and the control. The Mg concentrations in the maize shoots resulted in concentration levels between 3.5 and 5.5 mg/g DM, with the highest value (5.5 mg/g DM) after P-Salt treatment. Only a slight tendency to lower values was detected with the combination treatments, with higher P-Salt ratios (1:2) leading to higher Mg content.

*3.2. Effect of P-Fertilizers on Plant Nutrient Content*

For a comparison of the total net uptake of P, Ca and Mg per plant, the data in Figure 4 were re-evaluated as total content per plant (Figure 5). Similar to the dry matter results, the theoretical P content calculated for the combination treatments from the addition of the values of the single components was compared to the measured values to evaluate synergistic effects (material and methods: evaluation of synergistic effects Equations (3) and (4)).

Compared to the untreated controls, the total P-content per plant increased with all applied P fertilizers in the silty loam. Compared to TSP, combinations of the recycled fertilizers had similar or higher effects on P-content, with a slightly higher effect of SF(S) vs. SF (W). Differences between the application techniques were clearly visible for all fertilizer combination in favor of higher P-contents after dry application of the fractions. Furthermore, with dry application, the fertilizer ratio (1:2) caused higher P-contents compared to the ratio (1:1). Especially, the SF(S) + P-Salt (1:2) dry application showed highest total P-net-uptake per plant across all treatments. All combination treatments (SF + P-Salt), except for SF(W/S) + P-Salt (1:2) suspended, resulted in much higher P-contents than theoretically calculated from the single components' P-contents, indicating a synergistic effect between both fractions. The Ca-content per plant was increased by all fertilizer treatments compared to the control, with highest Ca-content for combination treatments, namely SF(W) + P-Salt (1:1) suspended and dry and the combination SF(S) + P-Salt (1:1)

suspended. Compared to TSP, the other fertilizer treatments resulted in similar or lower Ca- contents with the lowest Ca-content of the fertilizer treatments of P-Salt alone. The single solids resulted in slightly higher Ca-contents, with increased Ca-contents with SF(S) single treatment. Comparing the fertilizer treatments, the higher P-Salt ratio of 1:2 and the dry application technique had decreasing effects on Ca-content in those combinations with SF(W). For the SF(S) combinations, only decreasing effects on Ca-content with the dry application technique was observed. For Mg-content, all fertilizer combinations resulted in higher Mg-contents compared to the single fertilizer application and the reference TSP. Similar to Ca, the highest Mg- contents per plant were observed in the SF(W) + P-Salt (1:1) suspended and dry, whereas a higher P-Salt ratio (1:2) decreased the Mg-content. The SF(S) + P-Salt combinations resulted in slightly lower Mg-contents with no different effects between both ratios and different application techniques. No differences between single fertilizers P-Salt and SF(W) were observed, whereas SF(S) was slightly increased compared to SF(W). Compared to the reference TSP, all three single components resulted in lower Mg-contents.

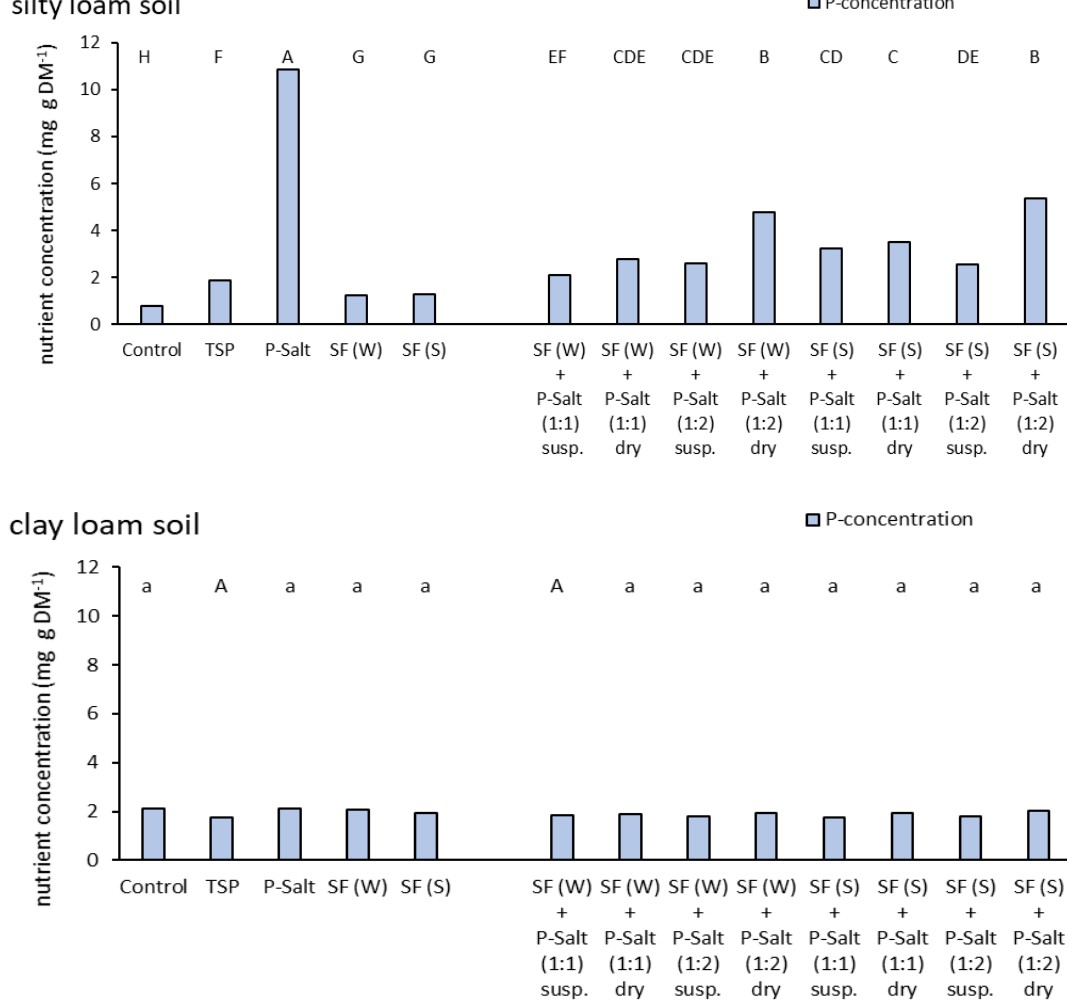

**Figure 4.** *Cont.*

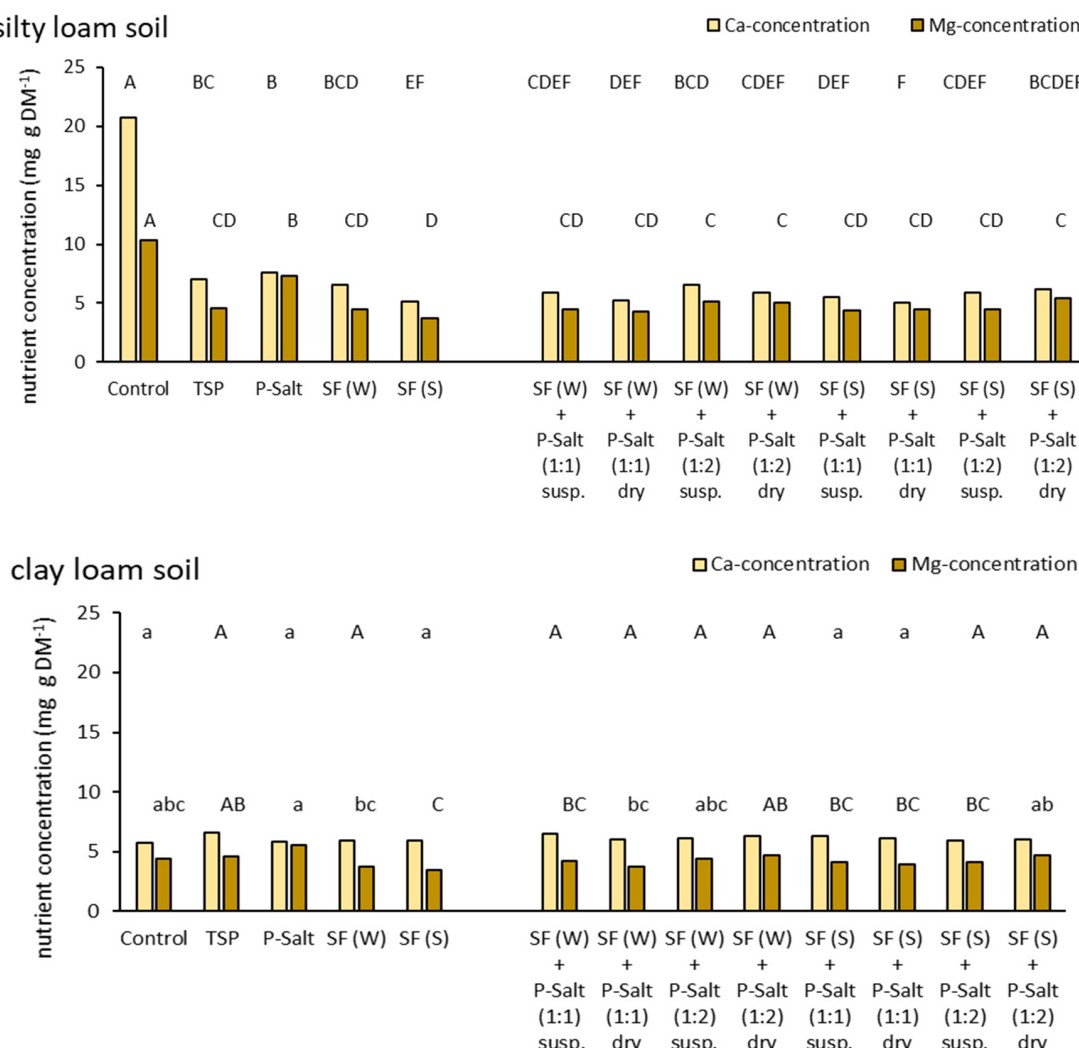

**Figure 4.** Maize P-, Ca- and Mg concentrations [mg g DM$^{-1}$] after soil treatment with different recycled digestate fertilizers in a total dose equivalent to 150 mg P (kg soil$^{-1}$). Please see Figure 3 for all other details. Data are means of four replicates each. Different letters denote significant differences (two-factorial Tukey's test with $p \leq 0.05$). Lowercase and uppercase letters indicate significantly different mean values between both soil types. Uppercase letters for both soil types indicate no significant difference.

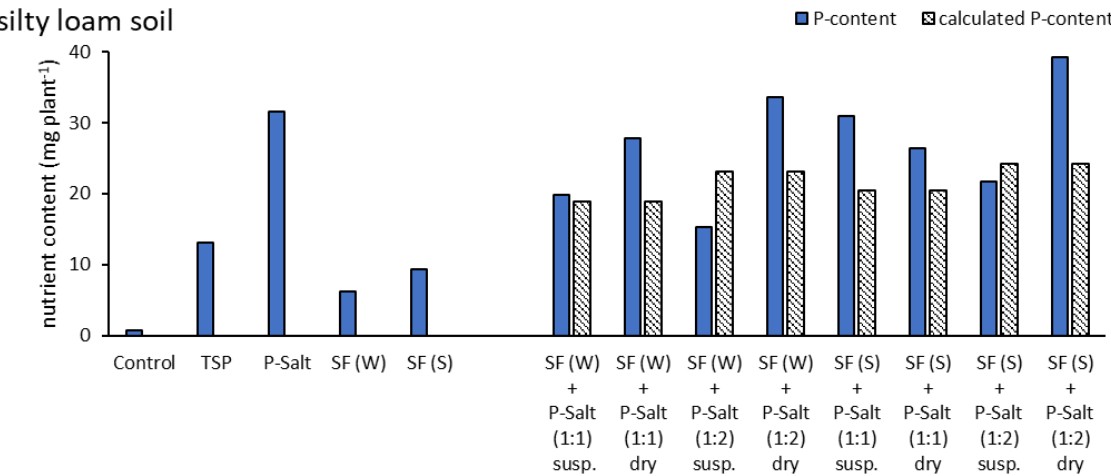

**Figure 5.** *Cont.*

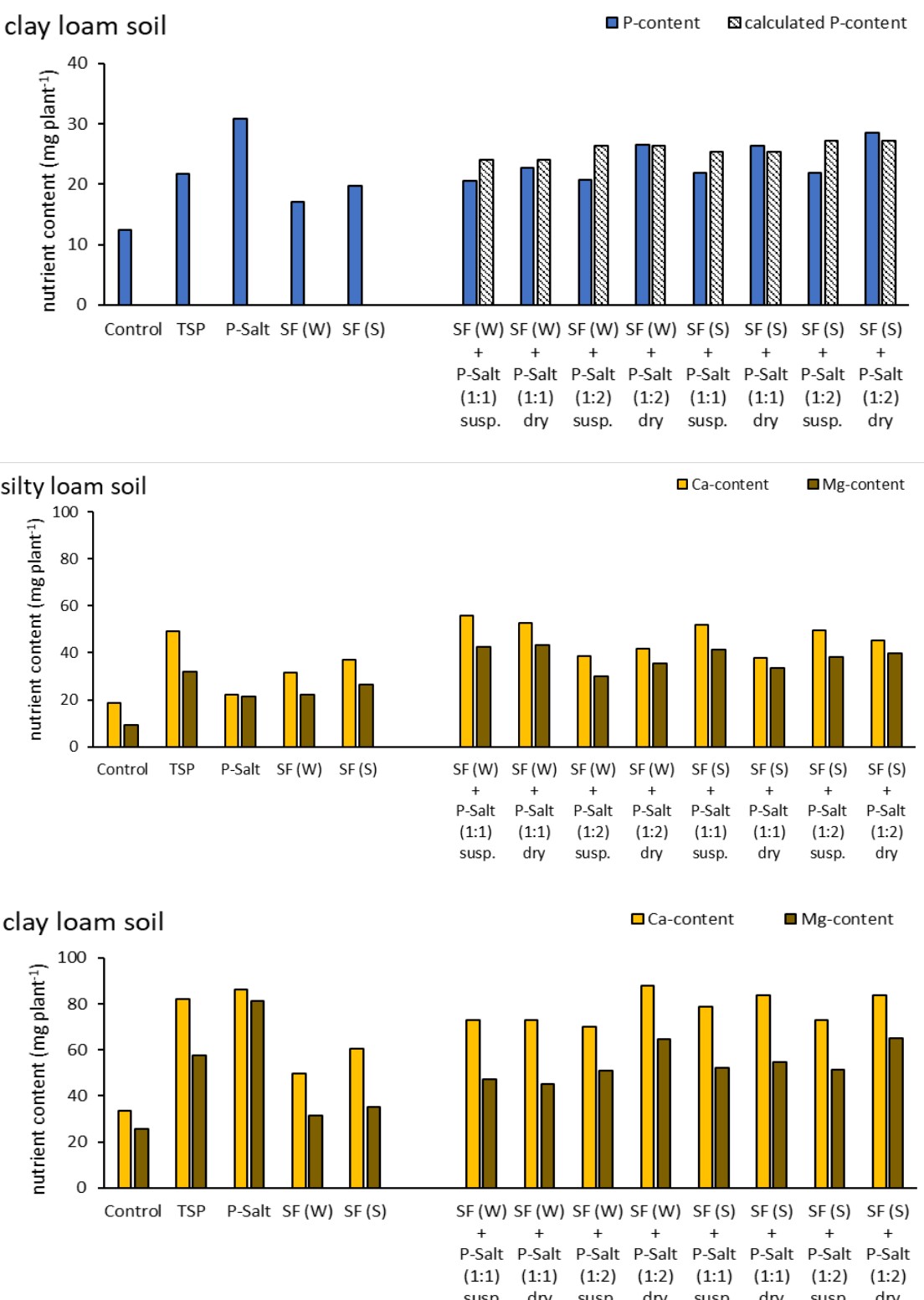

**Figure 5.** Total and calculated content per plant of P (P-content, calculated P content), and total content of Ca and Mg (Ca/Mg-content) in maize in mg/plant after soil treatment with different recycled digestate fertilizers in a total dose equivalent to 150 mg P (kg soil$^{-1}$). Please see Figure 3 for all other details. "Calculated P-content" represents a mathematical calculation of the combinations based on the results of the measured P-concentration of each single component (P-Salt, SF(W), SF(S)) to indicate possible additive effects (see material and methods: evaluation of synergistic effects, Equations (1) and (2)). Data are means of four replicates each.

Like in the silty loam, a single application of P-Salt resulted in the highest P-content per plant grown in the clay loam, and treatments with SF (W) or SF (S) alone were less effective but still at the level of TSP. Differences between the combination treatments were generally less pronounced and absolute P net uptake was mostly equal or higher compared to TSP, with a slight advantage of dry vs. suspended soil treatment. All combination treatments (SF + P-Salt), except for SF(S) + P-Salt (1:1) and (1:2) dry, resulted in similar or much lower P-contents than theoretically calculated from the single components' P-content, indicating no synergistic effect between both fractions. Ca-content was increased for all fertilizer treatments, with highest content with TSP, P-Salt and SF(W) + P-Salt (1:2) dry application. The other fertilizer combinations also resulted in increased Ca-contents compared to the single solids (SF(W/S)). Advantages of the combination treatments were observed with dry application technique, whereas different ratios had no effect. The single solid treatments resulted in increasing Ca-content for SF(S) compared to SF(W) treatment, with lowest Ca-contents with SF(W) compared to all other fertilizer treatments. Mg-content was the highest with P-Salt fertilizer followed both combination treatments SF(W/S) with higher P-Salt ratio (1:2) and dry application. Slightly increased Mg-contents were observed for SF(S) combination treatments compared to SF(W). Single solids (SF(W/S)) resulted in lowest Mg-contents, slightly higher for SF(S) compared to SF(W) and both significantly higher compared to the control.

### 3.3. Effect of P-Fertilizers on Plant Available P (CAL-P) in Soil

The CAL-P concentration in the soil solution, defined as the fraction being available to plants, was determined in parallel to the P-concentration in the plants in order to record the P-availability in the root zone as a function of the different fertilizer scenarios. Results are shown in Table 3.

**Table 3.** CAL-extractable P (CAL-P) in soil.

| P-Sources | Application Technique | CAL-P 2 Days after Fertilizer Incubation | CAL-P after Maize Harvest | CAL-P 2 Days after Fertilizer Incubation | CAL-P after Maize Harvest |
|---|---|---|---|---|---|
| | | Silty Loam [mg kg$^{-1}$] | Silty Loam [mg kg$^{-1}$] | Clay Loam [mg kg$^{-1}$] | Clay Loam [mg kg$^{-1}$] |
| Control | | 24 G | 15 F | 73 g | 54 e |
| TSP | | 61 F | 142 AB | 121 cde | 97 cd |
| P-Salt | | 205 A | 173 A | 151 abc | 139 a |
| SF(W) | | 97 E | 52 E | 78 fg | 91 d |
| SF(S) | | 68 F | 59 E | 94 efg | 101 bcd |
| SF(W)+P-Salt (1:1) | suspended | 113 CDE | 80 D | 123 BCDE | 115 abcd |
| | dry | 130 BCD | 123 BC | 103 def | 109 ABCD |
| SF(W)+P-Salt (1:2) | suspended | 101 DE | 125 BC | 117 CDE | 110 ABCD |
| | dry | 147 BC | 138 ABC | 141 ABC | 133 AB |
| SF(S)+P-Salt (1:1) | suspended | 104 DE | 104 CD | 161 ab | 105 ABCD |
| | dry | 145 BC | 129 ABC | 146 ABC | 122 ABCD |
| SF(S)+P-Salt (1:2) | suspended | 151 B | 119 BC | 167 A | 127 ABC |
| | dry | 148 BC | 139 ABC | 135 ABCD | 110 abcd |

CAL-extractable P (CAL-P) in mg kg$^{-1}$ soil cultivated with maize after soil treatment with different recycled digestate fractions in a total dose equivalent to 150 mg P (kg soil$^{-1}$); fractions were P-Salt (P-enriched precipitate from the liquid fraction), SF (W) (air-dried solid fraction), SF(S) (steam-dried solid fraction), and different combinations of them compared to unfertilized control and triple superphosphate (TSP); CAL-P was measured 2 days after fertilizer incubation prior to seeding and after maize harvest; please see Figure 1 for all other details. Data are means of four replicates each. Different letters denote significant differences (two-factorial Tukey's test with $p \leq 0.05$). Lowercase and uppercase letters indicate significantly different mean values between both different soil types. Uppercase letters for both soil type treatments indicate no significant difference.

All tested P-fertilizers significantly increased the free CAL-P concentration in the silty loam compared to the untreated control. This was observed directly after fertilization and after harvest of the maize plants. The CAL-P after fertilization (measured 2 days after dosing) was lowest in the untreated control (24 mg/kg soil) and highest in the soil fertilized with P-Salt (205 mg/kg soil). Compared to this, the reference TSP and the solid SF(S) had relatively low initial CAL-P concentrations with levels similar to the control. SF(W) resulted in slightly higher CAL-P compared to SF(S). The combination treatments resulted in higher CAL-P than the single solids alone, but lower than the P-Salt alone. The CAL-P was slightly increased with dry application technique, except for SF(S) + P-Salt (1:2). However, after harvest, the CAL-P in soil with TSP treatment had more than doubled. SF(W) and SF(S) fertilization showed, in contrast, higher initial P-concentrations that slightly decreased at harvest. By far, the highest initial and also terminal CAL-P-concentrations in soil were found after treatment with P-Salt alone. Combinations of fertilizer fractions lead to highest CAL-P-concentrations when P-Salt was applied dry in a ratio of 2:1 or 1:1, both leading to a level lower than P-Salt alone and the reference TSP. Significant differences in the application techniques (suspended or dry) were only noticed for SF(W) + P-Salt (1:2) and SF(S) + P-Salt (1:1) with higher CAL-P concentrations with the dry application technique. However, at harvest, the CAL-P concentrations in the soil had generally decreased, except with TSP which increased the final concentration a lot (up to 173 mg/kg soil). Interestingly, the combinations of P-Salt and SF resulted in a relatively constant available P level at harvest around or higher than 100 mg/kg soil. Significant differences in the application techniques (suspended or dry) were only noticed for SF(W) + P-Salt (1:1) again with higher CAL-P concentrations with the dry application technique.

In the clay loam, the tested P fertilizers showed higher P concentrations in the soil compared to untreated controls right after fertilization, except for (SF(W/S) that resulted in similar concentrations as the control of ca. 73/78/94 mg/kg soil. Highest CAL-P could be measured for P-Salt and the combination treatments SF(W) + P-Salt (1:2) dry, SF(S) + P-Salt for both ratios (1:1 and 1:2) and the suspended and dry application technique, especially after harvest for P-Salt alone. The solid SF(W) had low CAL-P after fertilization similar to the control, but after harvest CAL-P concentration was a bit increased to the level of TSP. Differences between SF(W) and SF(S) could be noticed after fertilization and after harvest with increased CAL-P with SF(S). Only slight CAL-P differences for the combinations of SF(W) + P-Salt after fertilization and after harvest were detected with increased CAL-P for the 1:2 ratio together with dry application. The combination with SF(S) + P-Salt resulted in higher CAL-P compared to the combinations with SF(W). The different application technique showed no significant effects.

## 4. Discussion

### 4.1. Effects of P-Fertilizers on Biomass Yield, Plant Nutrient Uptake, and CAL-P in Soil

P-enriched salt (P-Salt), recovered from the liquid fraction of a biogas plant digestate by alkaline precipitation, increased the dry matter (DM) yield of maize shoots, grown in a clay loam agricultural surface soil for 50 days. The increase was significant compared to a mineral reference fertilizer (TSP), dosed to the soil in P-equivalent, single pre-emergence amounts of 150 mg/kg. With respect to the absolute amount of P given in comparison with the minimum P content requirements defined by the German Fertilizer Ordinance [37] for different fertilizers (w/w content of P in Thomas phosphate 4.4%, superphosphate 7.0%, dicalcium-phosphate with Mg 8.7%), the precipitated P-Salt used in this study with a P content of 11% (w/w) was within that range [38]. The application of both fertilizers resulted in similar P concentrations in maize shoot dry matter. The absolute total P content after treatment with P-Salt was even higher than with TSP. Similar results were reported by Ehmann and Bach [39] in ornamental plants, and by Ehmann et al. [40] on spring barley and *Vicia faba* L. beans in different soils using P-Salt from pig manure recovered by the same P recycling process as in our study. Further studies from Vogel et al. [41], Lekfeldt et al. [42], Vaneeckhaute et al. [43], Cabeza et al. [1], Römer and Steingrobe [44], who had used other

recycled, P-enriched fertilizers from different sources, also showed that these products have high potential as P fertilizer. These findings indicate that P-enriched materials, derived from biological waste, can be effective P fertilizers that may reduce the use of mineral P sources. Likewise, struvite (ammonium magnesium phosphate) as a prominent example of recycled P from wastewater and sludge [45] was reported to perform as an effective slow release fertilizer [46].

The practical use as fertilizer, however, is highly dependent on the plant availability of the P components in the soil. P mobilization and uptake by soil microbes and plant roots play a vital role for crop yield [47]. In the presence of recycled P components, critical soil components may still be lacking in nutrient-poor soils, as can be seen in the results from the nutrient–depleted silty loam subsoil used in this study, where the effect of P-Salt on DM yield was lower than the reference TSP. An addition of the recycled solids (SF(W), SF(S)), however, resulted in similar DM yield compared to TSP. Obviously, the P-fertilization success depends on the soil/rhizosphere-plant continuum, as also mentioned, e.g., by Hinsinger [48] and Shen [8]. These authors reported that plants can make use of P fertilizers with low plant availability by changing the conditions in the rhizosphere e.g., through the change of pH or the release of phosphatases. An important mechanism of P release has also been attributed to mycorrhiza communities in the rhizosphere. The relationship between nutrient supply and enzyme activity is regulated by a negative (reciprocal) feedback mechanism [49]: when the nutrient supply is low, the enzyme activity is induced, and vice versa. The above may be an explanation why the addition of recycled digestate solids positively influenced the rhizosphere of the nutrient poor soil with respect to P mobilization and uptake by the plant. Possible mechanisms for this conditioning effect, defined as any ability to enhance crop yields and/or improve soil performance for any soil function, could be of physical (soil structure, gas exchange, water availability) or/and microbiological (soil microbiome, plant–microbe interactions) nature [50].

Triple superphosphate (TSP) is a mixture of calcium dihydrogen phosphate and monocalcium phosphate, $[Ca(H_2PO_4)_2 \times H_2O]$. A high proportion of P in TSP is directly water soluble, so that P is immediately available for plant uptake [1]. In contrast, the water solubility of P in P-Salt is very low. $NaHCO_3$ and $H_2SO_4$ can increase P solubility to 30–50% (Table 1). The exact mechanism of P mobilization from P-Salt is yet unknown. However, it is likely that processes in the rhizosphere, also facilitated by the dried solids of the digestate, are responsible for P mobilization. The slow but continuous release of P from P-Salt regulated in the rhizosphere may therefore represent a depot effect of this material in comparison to TSP, which, due to its high water solubility, might be subject to rapid immobilization in soil, making it rapidly unavailable to plant roots. The different results in the two soils tested in this experiment indicate that the recycled solids (SF(W), SF(S)) may act to enhance P-Salt utilization in soils that have low P content and insufficient P mobilization capability. Support for the hypothesis that the rhizosphere regulates P comes from our CAL-P measurements in the tested soils two days after fertilization and at harvest. Despite the differences in solubility, the measured CAL-P concentration after single P-Salt dosage and in all combinations with SF were equal or higher compared to TSP in both soils (both initially and after harvest).

A high correlation between plant DM yield and CAL-P in the agricultural clay loam soil was observed (Figure 3 vs. Table 3). Lowest DM and CAL-P were found in untreated controls, highest values after treatment with P-Salt. SF(S) alone was superior to SF(W) in both parameters. All combinations were comparable with TSP and only showed almost insignificant variability between each other. In the silty loam subsoil, correlation of both parameters was also given between the recycled solids (SF(S) > SF(W)). Like in the clay loam soil, differences between the combinations were generally low. The low effect on DM yield for P-Salt alone, however, was not correlated with low CAL-P in this soil. Here, DM yield was rather negatively correlated with the CAL-P. Likewise, in the mixtures, the negative effect of higher proportions of P-Salt on DM yield was rather inversely correlated with the CAL-P results. However, when compared to P concentrations in plant, the high

CAL-P for P-Salt alone and partly for the 1:2 combinations correlated well with the highest P-concentrations. Possible explanations for these two effects are: (1) that P-Salt contains a fraction that is easily water-soluble and available for immediate plant uptake and (2) that high P content in the plant may have an inhibitory effect on plant DM.

Support for hypothesis (1) comes from the finding that the measured concentrations of Mg and Ca in P-Salt fertilizer were high (Table 1), making it likely that a minor but significant portion of P in the P-Salt fraction is partly bound in Ca and Mg-Salts that are readily available in soil and easily taken up by plants [51,52], similar to TSP. Indeed, when comparing the Ca and Mg content in the maize shoots with DM yield, high Ca and Mg content in the plants mostly correlated with high DM yield in both soils. The fact, that the Mg and Ca status of maize plants is a reliable indicator for plant growth and yield was reported, e.g., by Potarzycki [53], Lecourieux et al. [54] and Szczepaniak et al. [55].

In summary, the results of this study show that P-Salt alone and combinations with solids recovered from a biogas digestate can be an effective fertilizer alternative to TSP in an agricultural surface soil typical for maize growth. Single doses of P-Salt at equimolar P-levels resulted in equivalent or higher values for three indicators of yield and nutrient supply, namely plant DM, P content and soil CAL-P concentration. In the silty loam subsoil, a model for a P depleted soil with low microbial activity, the P fertilization effects of P-Salt alone were also comparable to TSP regarding P content and soil CAL-P. In this situation, however, a mixture of P-Salt with the co-isolated solid fraction (SF) of the digestate was necessary for a plant DM yield comparable to TSP. A systematic analysis of synergistic effects between both recycled fractions is discussed below.

### 4.2. Fertilizer Effects of Different Fertilizer Combinations and Soil Application Techniques

Synergistic effects on DM yield were observed for combinations of P-Salt with different solid fractions of recycled biogas digestate (SF(W), SF(S)) mostly in the nutrient poor silty loam subsoil and less in the agricultural surface soil. This leads to the conclusion that the isolated solids might have a soil conditioning effect, especially on soils with low nutrient content by adding or stimulating soil microbial activity.

In a parallel study reported by Ehmann and Bach [39], recycled fertilizers produced identically to this study in the same biogas plant, were tested with sunflowers in a horticultural growth substrate in a greenhouse pot experiment. Synergistic effects of P-Salt and air-dried solids (SF(W)) on DM yield were also observed. Similar results were reported by Ehmann et al. [40] in greenhouse pot experiments. Other authors reported soil conditioning effects of recycled organic waste and biochars that enhanced the effect of organic fertilizers, improved soil quality [40,50,56] and decreased the need of inorganic fertilizers [57–60].

Regarding the combination of SF to P-Salt, a 1:1 ratio was almost as efficient as a ratio of 1:2 in the silty loam subsoil for DM yield, P concentration in plants and CAL-P. For the clay loam soil only slightly increased DM matter yields were observed for 1:2 ratios compared to 1:1, especially with dry application. Given that the concentration of P in the solids was considerably lower than in P-Salt, the results with combined fertilizers in the clay loam surface soil indicate the potential of the solid fractions to reduce the total amount of P-Salt needed, to be equally efficient as TSP. For practical use, the ratio between the recycled fertilizer components may therefore need to be optimized individually to the agricultural situation.

A comparison of the different application techniques revealed a slight advantage of dry vs. suspended application for DM yield increase and total P content in the plants. The low difference between both application techniques demonstrates that the process of adding the different fertilizer components to the soil is relatively robust for practical use. Suspension of the components may be more suitable in greenhouse horticulture, whereas dry application of solids would be more applicable on arable land.

### 4.3. Effects of Different Drying Procedures of the Solids (Air Dried vs. Steam Dried) on Fertilization

The treatment with the two different dried solid fractions - air-dried at 40 °C (SF(W)) and steam-dried at 120 °C (SF(S))—resulted in significant differences in DM yield, concentration and content of P, Mg and Ca in maize plants in both tested soils. Increased DM yield and nutrient content of P, Ca and Mg were observed with SF(S) treatment compared to SF(W) in both tested soils, whereas differences in CAL-P were less pronounced.

The chemical characterization of the recycled solids is shown in Table 1. In addition to an alkaline pH (8.5), SF(S) had only half the concentration of $H_2SO_4$—soluble P (sparingly plant available). SF(W) on the other hand, was shown to remain virtually unchanged during drying as shown by Awiszus et al. [25].

Different authors have reported that high-temperature drying processes of organic waste/material decrease the amount of organic P and increase the fraction of inorganic (more bio-available) P, a form of phosphorous that can be directly absorbed by plants (e.g., [61–64]). Despite the expectation that higher drying temperatures (here 120 °C for SF(S)) might negatively influence biological indicators of the fraction, a negative effect on plant growth could not be observed. One of the reasons might be the relatively high heat resistance of soil phosphatases, as reported by Eivazi et al. [65].

The process of superheated steam drying is well known in the foodstuff industry [66]. Superheated steam transfers its heat gently to the product to be dried and the water to be evaporated and, thus, acts both as heat source and as drying medium. The process results in much lower particle size and increased homogeneity of the material. This may contribute to the positive effects observed for SF(S) compared to SF(W).

### 5. Conclusions

The utilization of P recycled products as fertilizer is an important strategy to close nutrient cycles in agriculture and to save nutrient resources. The presented data demonstrate an obvious benefit of the use of P recycled fertilizers from a biogas digestate on agricultural soils. Our results show that effects were comparable to or even stronger than conventional triple superphosphate (TSP). Indicators for fertilizer efficacy in this study were (a) plant dry matter (DM) yield, (b) plant P concentration and content, (c) plant Ca and Mg concentration and content and (d) CAL-P in soil.

The ratio between the isolated fractions ("P-Salt", "solids") was decisive for the magnitude of effects on plant (DM), P concentration and absolute content in plants and P concentration in soil (CAL-P). In a nutrient-poor soil, synergistic effects between the fractions were observed, most likely due to an induced increase of the originally low microbial activity in the soil. Steam drying of the recycled solid fractions resulted in higher fertilizing effects compared to air-drying. Results of this greenhouse experiment indicate that recycled P-Salt and combinations with solids can be used as sustainable substitute for mineral P fertilizers.

The specific conditions in this study, namely soil, crop, and environmental conditions, may not be applicable to other specific agronomical situations. Furthermore, a more detailed analysis of the chemical composition of the recycled fertilizers would be helpful to understand effects and possible unwanted side effects of the material used. Dry matter as an indicator for crop yield is useful for energy crops like maize but may not reflect the yield situation for other crops at harvest.

Further research is needed to understand underlying processes and effects. Other combinations and ratios of the single products might cause different effects in different soils depending on biological processes through P mobilization and soil pH effects. Moreover, the P fertilizers were investigated in pot experiments only, and further confirmation is needed in field experiments over longer time scales (soil P processes are much slower compared to N processes) and for a wider range of soil types and cropping systems. Results would lead to a better agronomic management of recycled products as P fertilizers involving soil and rhizosphere processes and improving P-recycling efficiency in the future.

**Author Contributions:** I.-M.B. conceived, designed, and performed the experiments and analyzed the data. L.E. supported the experiments and analysis. I.-M.B. and T.M. are the lead authors and all authors contributed to paper writing. All authors have read and agreed to the published version of the manuscript.

**Funding:** This work was performed within the project "GOBi", which received funding from the Federal Ministry of Education and Research (BMBF) under grant agreement No. 03EK3525A.

**Data Availability Statement:** Data can be provided by the corresponding author.

**Acknowledgments:** We would like to thank Helene Ochott and Charlotte Haake for their support with the laboratory analyses and Ivan Guzman-Bustamante for statistical advice.

**Conflicts of Interest:** The authors declare no conflict of interest.

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
