# Peer review of "Efficiency of Recycled Biogas Digestates as Phosphorus Fertilizers for Maize"

_agriculture, doi:10.3390/agriculture11060553_

Round 1

Reviewer 1 Report

Dear editors of Agriculture,

many thanks for the opportunity to read a nice manuscript. My observations are summarized below:

  • Title. The title is nice, however, is might be good to make it more catchier. Probably a good idea nice be adding a key research question into the title. I would also suggest avoiding repetitions of occurring words. I also think that P-fertilization should be rather even the title (and the abstract) better explained to avoid every confusion (the journal is read wide a wider audience than just agricultural experts).
  • Abstract. I woudn´t use just "P" but rather explained in a laymen style. In this style the abstract should be rewritten as in its current version it seems to me too technical. I would avoid mentioning any abbreviations (even those obvious) in the abstract. I think that it would be good introductory sentences are more condensed, more space is devoted to methodology, data, experiment and more space (and clarity) is devoted to original results. Practicality of the results achieved should be more clearly commented.
  • Introduction. This section is good, however, more context would be useful here when the research problem is introduced. Please expand about the scale of the problem when P is overused etc. It seems to me that this part could be better structured and gradually start from more general introductions and target to concrete issues. Please consider substructuring into subsections. I think that the objectives (and hypotheses) could be better linked to the theory. Maybe, it might be good to have three sections of theory and from each one objective and hypothesis derived?
  • Methods and Materials. This part definitely needs to be better structured. The experiment is described, however more in-depth description would serve here well. Please consider adding a scheme where design of your research is shown. Such a scheme might add much clarity. Please focus on the descriptions of individual steps conducted. Please follow traditional marking of sections and subsection.
  • Table 1 seems to be in different fonts.
  • It would be good if more attention is devoted to the part on the Methods and Materials. Some part would better serve in theoretical part. Please visibly name the part on Methods and the part on Materials/Data. More graphics would certainly be good here.
  • When showing formulas, please add their numbering.
  • Please check and expand the part on statistical analyses.
  • Results. Please shorten the titles of subsections, these should be clearer. Figures certainly can be better prepared and more attractive (coloured?). I would not recommend naming figures within the picture, this doesn´t look too good.
  • Please check all the abreviations used if all are properly explained.
  • Results seem reasonable and nice. Section 3.1 and 3.2 are disproportially long, why not divide section 3.1 into two section?
  • Discussion. Plesae check subtitles (subsection?) and give them proper numbering as required in the guide for authors. It is a very good idea to have this part similarly structured (and numbered) as the results. We can see more interlinks then. Please develop more graphics in the manuscript, this is weaker part of the text.
  • Conclusion. Please add more about limitations of the study (Method, Experiment, Settings?, Data, Analyses) and expand the part with practical recommendations derived from the results. Please reflect here your hypotheses.

I think this this will be a nice fit for the journal, however, some work still has to be done. Let me thank the authors for the work they did so far and encourage them to make the revisions. This will be a nice paper when revised.

I hope the authors will find my comments useful.

I recommend a major revision.

Kind regards,

Reviewer 2 Report

The chapter Introduction is too extensive with unnecessary elements. such as from line 31 to line 39. Likewise, the elements from lines 44 to 69 and from 71 to 81 are unnecessary. These are elements that do not contribute much to the knowledge of the subject related to the publication, and are only statistical and popular data.

Reviewer 3 Report

Comment on manuscript No. agriculture-1227169: P-fertilization efficiency of recycled biogas digestates as fertilizers for maize

Manuscript No. agriculture-1227169 is an important study and has evaluated the impact of recycled P-fertilizers (biogas digestates) on maize production, plant biomass, P, Ca, and Mg concentration in shoots, and plant and available P in soil. 

Why maize plants were harvested at 50 days, kindly explain whether it was the complete duration of crop cycle or it has been done for the experiment only.

Authors are suggested to clarify the reason behind selecting the measurement parameters (shoot dry matter yield, P, Ca, and Mg concentration and total content in maize plants) , what about the other key macro and micronutrient?

The study could have been sounder if the authors can highlight the impact of the adopted amendment on grain yield and its nutritional quality as these are the key factors for agricultural productions.

Present conclusions are general, kindly provide to the points conclusions based on the studied parameters. You can recommend the best amendment over the others.

Round 2

Reviewer 1 Report

I have no further comments. The authors sufficiently responded to my observations and I believe that the quality of the paper has been significantly improved. 

I am very happy that I can support his paper.

Let me wish the authors all the best for their future work. It was my pleasure to collaborate on the development of this paper.

Kind regards,

Author Response

Dear Reviewer, thank you very much for your response and your kind words. I was very happy when I read that I met your expectations. Your comments were very helpful and it was a pleasure to improve the paper to the way it is now. Thank you very much again for your time and support. Kind regards